

# Molecular profiling and phenotypic evaluation of thermo-sensitive genic male sterility genes for high-yielding rice hybrids (*Oryza sativa* L.)

B. Nagendra Naidu[1], Manonmani Swaminathan[1], Nivedha Rakkimuthu[1], Pushpam Ramamoorthy[2], Kumaresan Dharmalingam[1], Raveendran Muthurajan[3], Selvi Duraisamy[4], Tushar Arun Mohanty[1] and Bonipas Antony John[1]

[1] Department of Rice, Tamil Nadu Agricultural University, Coimbatore, Tamil Nadu, India
[2] Department of Forage Crops, Tamil Nadu Agricultural University, Coimbatore, Tamil Nadu, India
[3] Directorate of Research, Tamil Nadu Agricultural University, Coimbatore, Tamil Nadu, India
[4] Department of SS&AC, Tamil Nadu Agricultural University, Coimbatore, Tamil Nadu, India

Corresponding author
Manonmani Swaminathan,
manonmanitnau@gmail.com

## ABSTRACT

Temperature-sensitive genic male sterility (TGMS) is crucial for boosting rice productivity and ensuring food security. In this study, we evaluated the morphological traits and genetic diversity of 57 rice TGMS lines under fertility-inducing (Gudalur) and sterility-inducing (Coimbatore) environments. Significant variations were observed in yield and floral characteristics, with flowering times ranging from 62.3 to 152.0 days. Lines such as TNAU 16S and TNAU 93S exhibited shorter plant heights and durations, along with higher numbers of productive tillers, making them promising candidates for hybrid breeding. Molecular profiling revealed that the *tms*8 gene was the most prevalent across the lines, with some carrying combinations of two or more TGMS genes. TNAU 38S and TNAU 60S possessed all four TGMS genes, ensuring stable sterility. These lines showed low fertility temperatures at Gudalur and low sterility temperatures at Coimbatore, indicating optimal conditions for hybrid seed production. Floral characteristics in lines like TNAU 19S, TNAU 126S-1 and TNAU 126S-2 were favorable, with total sterility under sterility-inducing conditions and increased fertility under fertility-inducing conditions, making them ideal for hybridization. For short-duration, semi-dwarf hybrids, TNAU 93S and TNAU 16S were identified as suitable female parents. Overall, this study highlights the significance of TGMS gene combinations for stable male sterility expression and identifies lines such as TNAU 37S, TNAU 60S and TNAU 85S as optimal for high-yielding two-line rice hybrids. The findings emphasize the potential for developing diverse, stable hybrids that can contribute to improved rice productivity and global food security.

## INTRODUCTION

Rice is a globally important crop, feeding over 50% of the world's population (*Muthayya et al., 2014*). Rice hybrids have been shown to increase yields by 20 to 30% (*Virmani, 1996*). Since 1976, large-scale commercial production of hybrid rice seeds in China has significantly boosted agricultural output, transforming rice cultivation and improving global food security. Hybrids are produced using either the three-line or two-line methods, which rely on cytoplasmic genic male sterility (CGMS) or environmental sensitive genic male sterility (EGMS), respectively. Of these, CGMS is more complex, requiring both maintainer (B) and restorer (R) lines, which limits the choice of parental varieties or elite lines. In contrast, the two-line method using thermo-sensitive genic male sterility (TGMS) is more cost-effective, as it eliminates the need for a maintainer line, simplifying seed multiplication and production. As a result, TGMS is considered a viable alternative to CMS-based three-line breeding due to its simplified hybrid seed production (*Shukla & Pandey, 2008*). Additionally, TGMS allows any fertile line to serve as a pollen parent, increasing the frequency of heterotic hybrids in TGMS-based hybrid breeding.

Characterizing TGMS lines for floral traits that influence seed setting and yield is crucial for selecting suitable parental lines in hybrid breeding. Generally, TGMS lines become sterile when temperatures exceed 25–30 °C during panicle initiation and anthesis and revert to fertility when temperatures drop below 25–30 °C. Low temperatures trigger the development of non-pollen uninucleate cells (*Peng et al., 2009*). Several TGMS-related genes have been mapped, including *tms*1 (*Wang et al., 1995*), *tms*2 (*Pitnjam et al., 2008*), *tms*3 (*Subudhi et al., 1997*; *Lang et al., 1999*), *tms*4 (*Dong et al., 2000*), *tms*5 (*Wang et al., 2003*; *Nas et al., 2005*; *Jiang et al., 2006*; *Yang et al., 2007*), *tms*6 (*Lee, Chen & Suh, 2005*), *tms*7(t) (*Li RongBai, Pandey & Pankaj Sharma, 2005*), *tms*8 (*Hussain et al., 2012*), *tms*9 (*Sheng et al., 2013*), *tms*9-1 (*Qi et al., 2014*), and *tms*X (*Peng et al., 2010*). The single-gene recessiveness of TGMS in classical breeding requires monitoring sterility and seed production across various environments, thereby extending the breeding cycle (*Borkakati & Virmani, 1996*). The stability of TGMS lines is influenced by the presence and expression of specific male sterility genes, as well as their interaction with environmental factors. In some cases, TGMS lines are controlled by a single major gene, such as *tms*5, which causes temperature-sensitive male sterility. This simplifies stability management, as the sterility trait is primarily controlled by one gene. However, other TGMS lines may involve multiple genes or genetic loci contributing to male sterility, increasing the complexity of stability management. Complex interactions between multiple genes can lead to efficient expression patterns and potentially enhance stability.

Due to extensive mapping and cloning efforts, a wide range of PCR-based markers have been developed to screen and identify different TGMS genes. DNA markers closely linked to TGMS genes that cause complete sterility at specific temperatures can be used for efficient marker-assisted selection (MAS), which is faster than traditional pollen sterility tests. SSR markers can be used to assess the uniformity and stability of TGMS lines across generations and environmental conditions. Despite advancements in TGMS line development, molecular profiling had not been conducted previously in Tamil Nadu

Agricultural University (TNAU). This research aims to address that gap by providing molecular insights, contributing to a deeper understanding of TGMS traits to support sustainable rice breeding and adaptation to diverse environmental conditions. This study focuses on characterizing TGMS lines for sterility-related genes, yield and floral traits to identify the best-performing and most stable TGMS lines for developing high-yielding two-line hybrids.

## MATERIALS & METHODS

This study focused on the evaluation of 57 newly developed Temperature-Sensitive Genic Male Sterility (TGMS) rice lines (Table 1), developed through selection, mutation and pedigree breeding at TNAU, Coimbatore. The TGMS lines were evaluated under two contrasting environments. The sterility-inducing environment at TNAU, Coimbatore (11.0°N, 77.0°E, elevation 426.72 m) featured maximum temperatures exceeding 31 °C, while the fertility-inducing environment at the Hybrid Rice Evaluation Centre (HREC), Gudalur (11.5°N, 76.5°E, elevation 1,300 m) had maximum temperatures below 18 °C (Fig. 1). These distinct conditions facilitated the optimal expression of TGMS traits for evaluation.

### Observations recorded

One-month-old seedlings were transplanted in a randomized block design (RBD) with three replications during winter 2022 at TNAU and monsoon 2023 at HREC, with a spacing of 20 × 20 cm. Fertilizer application was carried out at the rate of 150:60:60 kg/ha of NPK, with nitrogen and potassium applied in three split doses during the basal, active tillering and panicle initiation stages, adhering to the recommended field management guidelines from the TNAU Agritech portal (http://www.agritech.tnau.ac.in/). Yield-related traits such as days to 50% flowering, plant height, productive tillers, panicle length, pollen fertility, spikelet fertility, and single plant yield were recorded in the fertility-inducing environment (HREC). Floral traits, including pollen sterility, stigma length, stigma exertion and glume angle, were measured under sterility-inducing conditions (TNAU). Pollen fertility was assessed by staining mature anthers from five plants with a 1% potassium iodide solution. Fertile pollen appeared dark blue under the light microscope, while sterile pollen remained colourless. Pollen fertility percentage was calculated using the formula:

$$\text{Pollen sterility percentage} = \frac{\text{Number of sterile pollens}}{\text{Total number of pollens}} \times 100.$$

The environmental conditions were carefully monitored to capture the natural transition between fertility and sterility. Minimum and maximum temperatures at critical stages were documented following the protocol established by *Vinodhini et al. (2019)*. TGMS lines were exposed to varying environmental conditions, ranging from 13 °C to 38 °C, reflecting the spectrum of natural cycles. The average minimum and maximum temperatures at which the shift from fertility to sterility occurred were defined as the critical fertility temperature (CFT) and critical sterility temperature (CST), respectively. Observations were taken from five randomly selected plants per genotype, ensuring representative data. These values were
**Table 1  List of temperature sensitive genic male sterile lines with parentage.**

| S. No. | TGMS line | Pedigree |
|---|---|---|
| 1 | TNAU 1S | Selection from CBDHTS 0235 |
| 2 | TNAU 2S | Selection from CBTS 0282 (TS29/IR68281B-400) |
| 3 | TNAU 4S | Selection from GD 98049-1-1(CBGD0502) |
| 4 | TNAU 4S-1 | Selection from GD 98049-1-1(CBGD0502) |
| 5 | TNAU 15S | Selection from IR 73827–23 S |
| 6 | TNAU 16S | Selection from IR 73824 S |
| 7 | TNAU 18S | Selection from IR 75589-41-13-17-15-22S |
| 8 | TNAU 19S | Selection from IR 75589-41-13-17-15-3S |
| 9 | TNAU 23S | TS 16 x IR 36-1-2-122 |
| 10 | TNAU 30S | TS 29 x IR682818-400-4-2-3-1 |
| 11 | TNAU 31S | TS 16 x MRST 9-292-2-1 |
| 12 | TNAU 34S | TS 29 x IR 68281B-400-2-6 |
| 13 | TNAU 37S | TS 29 x IR 68281B-400-2-1 |
| 14 | TNAU 38S | TS 29 x IR 68281B-25-5-2 |
| 15 | TNAU 39S | TS 29 x IR 68281B-400-2-3-2 |
| 16 | TNAU 45S | Selection from GD 99017-17 |
| 17 | TNAU 50S | Selection from GD 99017-10 |
| 18 | TNAU 51S | Selection from GD 98049-1-49 |
| 19 | TNAU 53S | Selection from GD 98049-29 |
| 20 | TNAU 59S-1 | Mutant from ADT 39 at Thenkasi |
| 21 | TNAU 59S-2 | Mutant from ADT 39 at Thenkasi |
| 22 | TNAU 60S | Spontaneous mutant from PMK 3 (Pet Type) |
| 23 | TNAU 71S | Selection from DRR 29S |
| 24 | TNAU 82S | Selection from GDR 9 S |
| 25 | TNAU 83S | TNAU 45S-2 |
| 26 | TNAU 85S | TS 29 100GY |
| 27 | TNAU 86S | Spontaneous mutant MS 3 |
| 28 | TNAU 92S | Selection from DRR 23 S |
| 29 | TNAU 93S | Selection from DRR 28 S |
| 30 | TNAU 95S | Selection from TNAU 65 S (TS 29/IR 62917-2-3-2-1-1) |
| 31 | TNAU 98S | GD98049-3-29-1 |
| 32 | TNAU 100S | ADT 39 100GY |
| 33 | TNAU 101S | DRR 23S-1 |
| 34 | TNAU 103S | CBTS0282-27-4-15-15 |
| 35 | TNAU 106S | CBTS0282-27-4-12-15 |
| 36 | TNAU 107S | GD98029-29 |
| 37 | TNAU 111S | TS29150GY-3 |
| 38 | TNAU 112S | MS4/TNAU51-1 |
| 39 | TNAU 113S | TNAU 4S-7-2/BPT5204 |
| 40 | TNAU 114S | TNAU 4S-2/BPT 5204 |
| 41 | TNAU 115S | TS 06-207-1-28 |
| 42 | TNAU 115S-1 | TS 06-207-1-28 |

**Table 1** (*continued*)

| S. No. | TGMS line | Pedigree |
|---|---|---|
| 43 | TNAU 116S | CBTS 282-6-22-1 |
| 44 | TNAU 120S | CBTS 0282-282-7 |
| 45 | TNAU 126S-1 | TNAU 15S/Sona Mashuri -2 |
| 46 | TNAU 126S-2 | TNAU 15S/Sona Mashuri -2 |
| 47 | TNAU 127S | TNAU 15S/Sona Mashuri -3 |
| 48 | TNAU 129S | MS 2S-2 |
| 49 | TNAU 131S | CO 49/TS 29 (F4 27S) |
| 50 | TNAU 132S | CO 49/TS 29 (F4 19S-1) |
| 51 | TNAU 135S | TS29/CO 49 (F4 17S) |
| 52 | TNAU 136S | TS 06-182-2 |
| 53 | TNAU 137S-1 | TNAU 4S-1-2/CB06-564 |
| 54 | TNAU 137S-2 | TNAU 4S-1-2/CB06-564 |
| 55 | TNAU 142S | TNAU 19S-1 |
| 56 | TNAU 143S | TNAU 19S-2 |

then averaged to provide a comprehensive understanding of the response of TGMS lines under dynamic environmental conditions, bridging the relationship between temperature sensitivity and the expression of sterility and fertility traits.

## DNA extraction

Two-week-old seedlings from each line were selected from the nursery and young leaves were collected for DNA extraction using the protocol given by *Doyle & Doyle (1987)*. The plant tissues were flash-frozen in liquid nitrogen to preserve their integrity and ground into a fine powder using a mortar and pestle. The pulverized tissue was transferred into a tube containing an extraction buffer (CTAB) and incubated at 65 °C for 30 min with periodic gentle mixing to ensure thorough processing. Following incubation, an equal volume of chloroform: isoamyl alcohol (24:1) was added to the mixture, which was then gently mixed and centrifuged at 12,000 rpm for 10 min at room temperature. The aqueous phase, containing the extracted DNA, was carefully transferred to a fresh tube. DNA was precipitated by adding 0.6 volumes of isopropanol or 2 volumes of 100% ethanol, followed by incubation at −20 °C for a minimum of one hour or overnight to encourage precipitation. The DNA was pelleted by centrifugation at 12,000 rpm for 10 min. The supernatant was discarded, and the pellet was washed with 70% ethanol and centrifuged again for 5 min at the same speed. The DNA pellet was allowed to air dry for 10–15 min before being dissolved in TE buffer or a similar solution. The dissolved DNA was stored at 4 °C, typically overnight, to ensure complete dissolution. RNA contamination was removed by treating the solution with RNase A. The quality and concentration of the extracted DNA were evaluated using a spectrophotometer. Absorbance readings were taken to confirm the purity, with an optimal 260/280 ratio of ∼1.8 indicating high-quality DNA suitable for downstream analyses.

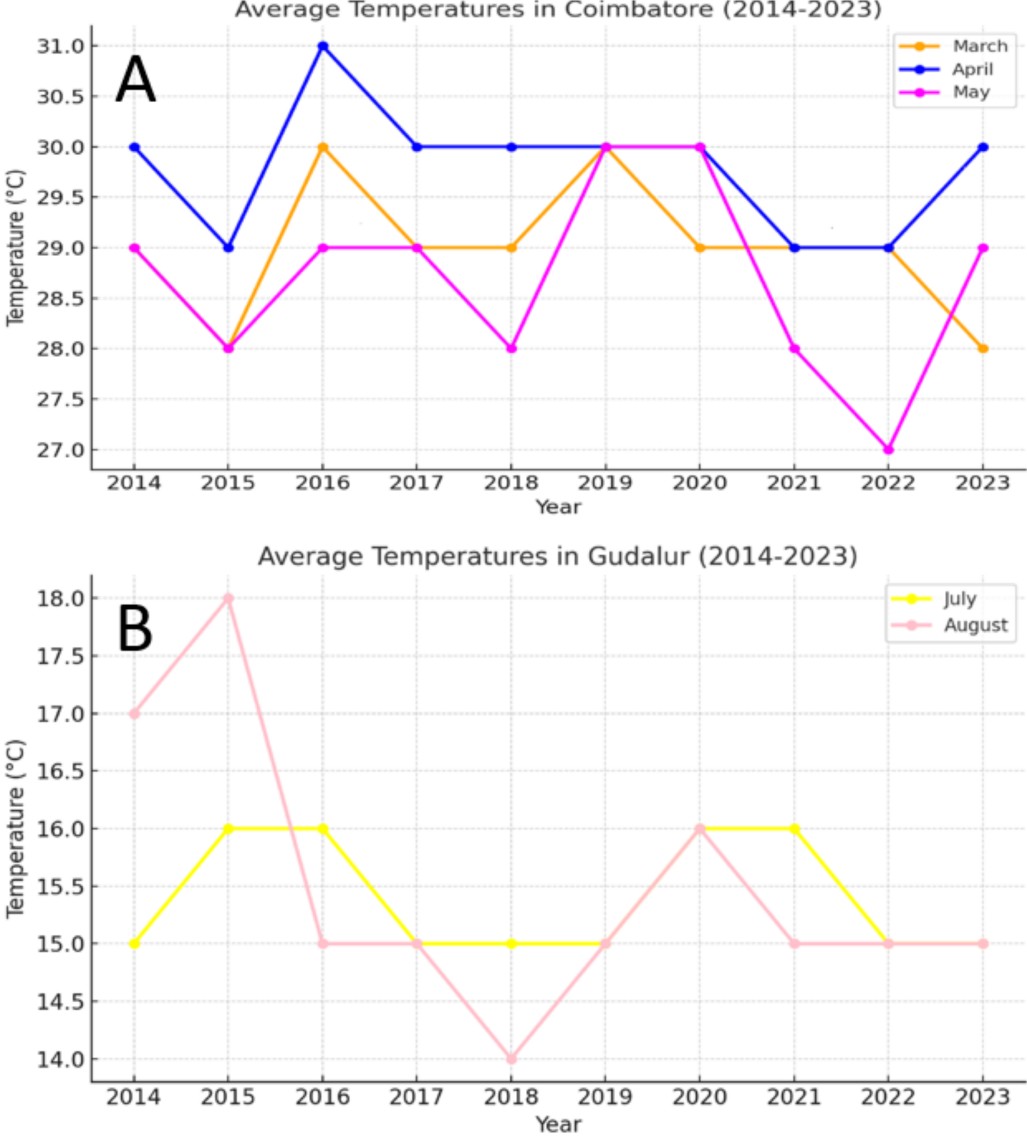

**Figure 1** **(A) Weather data of sterility inducing location (Coimbatore), (B) weather data of fertility-inducing location (Gudalur).** (A) The average temperature recorded during the month of April to May is 27 °C to 31 °C. This temperature is highly favorable for inducing sterility in the TGMS lines. (B) Temperature range of 14 °C to 18 °C for the past ten years showed that these temperature range is highly favorable for inducing pollen fertility.

## Molecular assay

The polymerase chain reaction (PCR) was performed using eight SSR primers (Table 2) specific to the *tms4, tms5, tms8* and *tms10* genes. Each reaction was carried out in a 10 μl mixture containing 2 μl of template DNA (20 ng/μl), 0.5 μl of forward primer (10 μM), 0.5 μl of reverse primer (10 μM), 4 μl of 2X master mix and 3 μl of sterile water. The PCR thermal cycling profile included an initial denaturation step at 95 °C for 5 min, followed by 35 cycles of denaturation at 94 °C for 1 min, annealing at 55 °C for 45 s, and extension at

**Table 2   List of SSR markers used for molecular profiling of TGMS lines.**

| S. No. | Marker | Chromosome locus | Sequence | LD (cM) | AT (°C) | APS (bp) | SSR motif | Gene | References |
|---|---|---|---|---|---|---|---|---|---|
| 1. | RM27 | 2 | F: TTTTCCTTCTCACCCACTTCA R: TGCTATAAAAGGCATTCGGG | 1.75 | 55 | 158 | (GA)7 | tms4 | *Dong et al. (2000)* |
| 2. | RM174 | 2 | F:AGCGACGCCAAGACAAGTCGGG R:TCCACGTCGATCGACACGACGG | 1.32 | 55 | 208 | (AGG)7(GA)10 | tms5 | *Wang et al. (2003)* |
| | RM5897 | 2 | F: GGCATCTTCCCCTCTCTCTC R: CCAACCCAAACCAGTCTACC | 1.52 | 55 | 141 | (ATT)15 | tms5 | *Wang et al. (2003)* |
| | RM6247 | 2 | F:CGCTCTTGTCTTTACTCCCG R: GCTGCTGCTGCTTCTTTTC | 3.29 | 55 | 103 | (CTC)8 | tms5 | *Wang et al. (2003)* |
| | RM7575 | 2 | F: GGTTTGATCTCGCGTCTCTC R: CCAGCAGCGAGAGAGATAG | 2.01 | 55 | 118 | (TCTA)8 | tms5 | *Wang et al. (2003)* |
| 3. | RM21 | 11 | F: ACAGTATTCCGTAGGCACGG R: GCTCCATGAGGGTGGTAGAG | 4.3 | 55 | 157 | (GA)18 | tms8 | *Hussain et al. (2012)* |
| | RM224 | 11 | F: ATCGATCGATCTTCACGAGG R: TGCTATAAAAGGCATTCGGG | 1.12 | 55 | 157 | (AAG)8(AG)13 | tms8 | *Hussain et al. (2012)* |
| 4. | RM300 | 2 | F: GAAACGTGTAGCAGTACGCC R:ACCAAGCTCTTCATCAACGG | 2.65 | 55 | 121 | (GTT) | tms10 | *Peng et al. (2010)* |

**Notes.**

LD, Linkage distance (cM); AT, Annealing temperature (°C); APS, Amplified Product Size (bp).

72 °C for 30 s. A final extension step was performed at 72 °C for 10 min. Amplified samples were stored at 4 °C until further analysis (*Nivedha et al., 2024*). The PCR products were separated on a 3% agarose gel prepared in 1X TBE buffer, alongside a 100 bp DNA ladder (Bio-Helix) for size reference. The gel was visualized using Bio-Rad imaging equipment under UV trans illumination, enabling clear differentiation of the amplified fragments.

## Statistical analysis methodology

Statistical analyses were performed using R software to ensure rigorous evaluation and enhance the accuracy of the results. The "aov" package was utilized for analysis of variance (ANOVA), and the "cor" package was employed for correlation analysis. To calculate descriptive statistics, including the mean and standard error of the mean (SEM), the psych package was used in R-Studio 4.3.2.

## RESULTS

### Morphological characterization

A detailed evaluation of 57 TGMS rice lines was undertaken to profile the genes governing male sterility and to explore their associations with floral traits that enhance outcrossing potential, an essential element in optimizing hybrid seed production. Significant genotypic variation in yield and floral characteristics was observed through analysis of variance (Supplemental Information 1). Traits sensitive to temperature changes, including pollen sterility, stigma length, stigma exertion and glume angle, were evaluated under sterility-inducing conditions at the Paddy Breeding Station, Coimbatore, while agro-morphological traits were recorded at the Hybrid Rice Evaluation Centre (HREC), Gudalur, under fertility-inducing conditions.

The TGMS lines exhibited flowering durations ranging from 62.3 to 152.0 days (Supplemental Information 2). Among short-duration genotypes, TNAU 112S (62.3 days), TNAU 15S (72.3 days), TNAU 39S (86.0 days), TNAU 16S (91.0 days) and TNAU 93S (91.0 days) showed the earliest flowering. Plant heights varied between environments, ranging from dwarf to tall, with TNAU 16S and TNAU 93S recording the shortest height (66.67 cm). Productive tiller counts were highest in TNAU 16S (27.67), TNAU 95S (25.00) and TNAU 19S (22.33), while TNAU 132S exhibited the fewest productive tillers (10.0). All TGMS lines demonstrated panicle exertion greater than 80%, with TNAU 143S (106.35%), TNAU 83S (103.45%) and TNAU 82S (100.00%) achieving the highest values. TNAU 103S recorded the longest panicle length (26.33 cm), while TNAU 93S had the shortest (15.00 cm). Grain counts ranged from 34.0 in TNAU 112S to 326.0 in TNAU 114S. The highest single-plant yields were observed in TNAU 114S (43.0 g), TNAU 131S (42.3 g) and TNAU 34S (41.0 g), whereas TNAU 101S had the lowest yield (7.7 g). Temperature sensitivity was critical for regulating sterility and fertility transitions. The critical fertility temperature (CFT) at Gudalur ranged from 15.5 °C to 19.0 °C, while the critical sterility temperature (CST) at Coimbatore ranged from 26.5 °C to 31.0 °C (Supplemental Information 3, Fig. 2). During sterility-inducing conditions, most accessions displayed complete pollen sterility (Table 3, Fig. 3). Under fertility-inducing conditions, pollen fertility ranged from 31.9% in TNAU 93S to 98.6% in TNAU 95S, with TNAU 16S (97.1%) and TNAU 19S (97.0%) also demonstrating high fertility. Conversely, poor fertility was observed in TNAU 112S (50.6%) and TNAU 111S (50.8%). High spikelet fertility was noted in TNAU 15S (86.1%), TNAU 126S-1 (84.4%) and TNAU 16S (82.9%), while low fertility was observed in TNAU 113S (14.7%), TNAU 93S (16.6%) and TNAU 31S (24.8%).

Floral traits measured under sterility-inducing conditions revealed stigma lengths ranging from 1.2 mm in TNAU 71S to 2.48 mm in TNAU 113S. Additional lines with significant stigma lengths included TNAU 112S (2.44 mm), TNAU 100S (2.32 mm) and TNAU 4S (2.22 mm) (Fig. 4). Glume angles varied from 10.0° in TNAU 135S to 31.7° in TNAU 126S-1 and TNAU 126S-2, with notable values also observed in TNAU 2S (30.0°) and TNAU 16S (28.3°). Stigma exertion, a key trait for enhancing outcrossing potential, was highest in TNAU 112S (85.9%), TNAU 92S (84.6%) and TNAU 142S (76.6%), while the lowest values were recorded in TNAU 98S (10.9%), TNAU 46S (14.8%) and TNAU 30S (19.3%) (Fig. 5).

## Allelic diversity of rice TGMS genes

The molecular profiling of TGMS lines identified the presence or absence of four major rice TGMS genes *tms4*, *tms5*, *tms8* and *tms10* using SSR markers. Detailed results are presented in Table S1, and the electrophoresis patterns of SSR markers linked to these genes are shown in Fig. 6. The genetic frequencies of the four TGMS genes ranged from 8.92% to 69.64%. Among these, *tms8* was the most widely distributed, occurring in 69.64% of the genotypes, followed by *tms10* (57.14%), *tms5* (16.07%) and *tms4* (8.92%). The rice marker RM27 (158 bp) linked to the *tms4* gene was detected in five of the 57 accessions. Nine accessions exhibited amplification for RM174 (208 bp), RM5897 (141 bp), RM6247 (103 bp) and RM7575 (118 bp), markers associated with *tms5*. Positive bands for RM21 (157
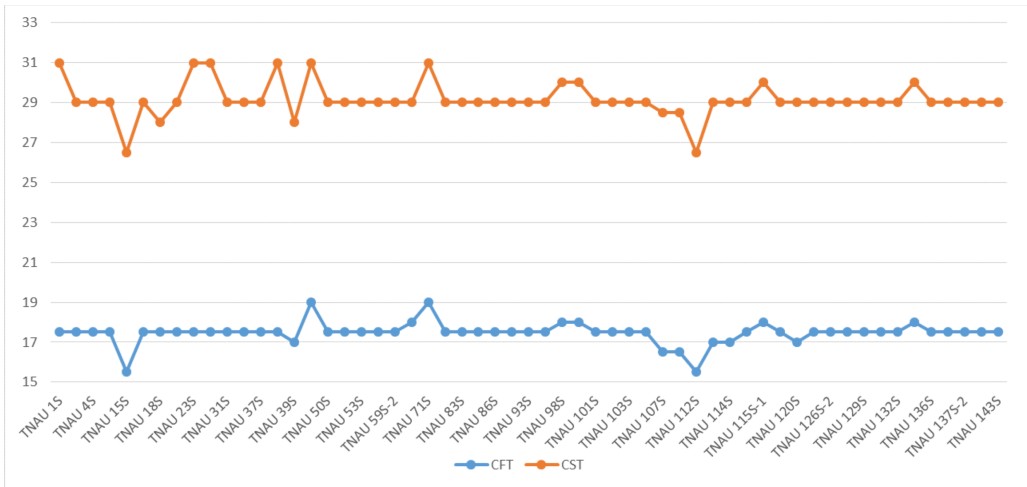

**Figure 2** **Critical fertility- sterility temperatures of different TGMS lines.** CFT, Critical fertility temperature; CST, Critical sterility temperature.

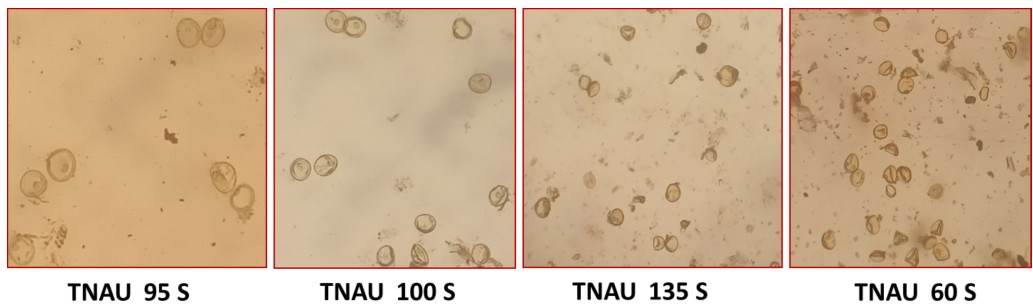

| TNAU 95 S | TNAU 100 S | TNAU 135 S | TNAU 60 S |

**Figure 3** **Pollen sterility in promising TGMS lines.**

bp) and RM224 (157 bp), markers connected to *tms8*, were observed in 39 genotypes. The marker RM300 (121 bp), associated with *tms10*, was amplified in 32 accessions.

Two accessions, TNAU 38S and TNAU 60S, demonstrated the presence of all four TGMS genes, showing positive amplification for all associated markers. Six accessions, including TNAU 23S, TNAU 39S, TNAU 59S-1, TNAU 82S, TNAU 83S and TNAU 85S, carried three TGMS genes. Additionally, 22 lines possessed two TGMS genes, 26 lines carried three genes, and 12 lines had only one gene, either *tms8* or *tms10*. These findings illustrate the diverse genetic composition of TGMS lines, with varying combinations of major TGMS genes contributing to their sterility characteristics. The presence of multiple genes in some accessions, particularly TNAU 38S and TNAU 60S, highlights their potential for use in breeding programs aimed at developing stable and high-performing rice hybrids.

### Genetic diversity between the *tms* genes

Genotyping of TGMS lines for the *tms4, tms5* and *tms10* genes on chromosome 2, and *tms8* on chromosome 11, revealed seven distinct gene combinations (Table 4). The combination

**Table 3  Critical sterility and fertility temperature of TGMS lines in both sterility and fertility permitting conditions.**

| S. NO. | Entry | DFF | DFBD | DPMC | MIN–MAX GDL | MIN–MAX CBE | CFT | CST |
|---|---|---|---|---|---|---|---|---|
| 1 | TNAU 1S | 115 | 90 | 103 | 14–22 | 24–38 | 17.5 | 31 |
| 2 | TNAU 2S | 97 | 72 | 85 | 11–24 | 20–38 | 17.5 | 29 |
| 3 | TNAU 4S | 94 | 69 | 82 | 11–24 | 20–38 | 17.5 | 29 |
| 4 | TNAU 4S-1 | 99 | 74 | 87 | 11–24 | 20–38 | 17.5 | 29 |
| 5 | TNAU 15S | 72 | 47 | 60 | 8–23 | 17–36 | 15.5 | 26.5 |
| 6 | TNAU 16S | 91 | 66 | 79 | 11–24 | 20–38 | 17.5 | 29 |
| 7 | TNAU 18S | 94 | 69 | 82 | 11–24 | 20–36 | 17.5 | 28 |
| 8 | TNAU 19S | 108 | 83 | 96 | 11–24 | 20–38 | 17.5 | 29 |
| 9 | TNAU 23S | 114 | 89 | 102 | 14–22 | 24–38 | 17.5 | 31 |
| 10 | TNAU 30S | 117 | 92 | 105 | 14–22 | 24–38 | 17.5 | 31 |
| 11 | TNAU 31S | 101 | 76 | 89 | 11–24 | 20–38 | 17.5 | 29 |
| 12 | TNAU 34S | 101 | 76 | 89 | 11–24 | 20–38 | 17.5 | 29 |
| 13 | TNAU 37S | 97 | 72 | 85 | 11–24 | 20–38 | 17.5 | 29 |
| 14 | TNAU 38S | 114 | 89 | 102 | 14–23 | 24–38 | 17.5 | 31 |
| 15 | TNAU 39S | 86 | 61 | 74 | 11–23 | 20–36 | 17 | 28 |
| 16 | TNAU 45S | 114 | 89 | 102 | 14–24 | 24–38 | 19 | 31 |
| 17 | TNAU 50S | 108 | 83 | 96 | 11–24 | 20–38 | 17.5 | 29 |
| 18 | TNAU 51S | 108 | 83 | 96 | 11–24 | 20–38 | 17.5 | 29 |
| 19 | TNAU 53S | 112 | 87 | 100 | 11–24 | 20–38 | 17.5 | 29 |
| 20 | TNAU 59S-1 | 109 | 84 | 97 | 11–24 | 20–38 | 17.5 | 29 |
| 21 | TNAU 59S-2 | 107 | 82 | 95 | 11–24 | 20–38 | 17.5 | 29 |
| 22 | TNAU 60S | 138 | 113 | 126 | 12–24 | 20–38 | 18 | 29 |
| 23 | TNAU 71S | 116 | 91 | 104 | 14–24 | 24–38 | 19 | 31 |
| 24 | TNAU 82S | 111 | 86 | 99 | 11–24 | 20–38 | 17.5 | 29 |
| 25 | TNAU 83S | 108 | 83 | 96 | 11–24 | 20–38 | 17.5 | 29 |
| 26 | TNAU 85S | 101 | 76 | 89 | 11–24 | 20–38 | 17.5 | 29 |
| 27 | TNAU 86S | 108 | 83 | 96 | 11–24 | 20–38 | 17.5 | 29 |
| 28 | TNAU 92S | 96 | 71 | 84 | 11–24 | 20–38 | 17.5 | 29 |
| 29 | TNAU 93S | 91 | 66 | 79 | 11–24 | 20–38 | 17.5 | 29 |
| 30 | TNAU 95S | 112 | 87 | 100 | 11–24 | 20–38 | 17.5 | 29 |
| 31 | TNAU 98S | 124 | 99 | 112 | 12–24 | 22–38 | 18 | 30 |
| 32 | TNAU 100S | 126 | 101 | 114 | 12–24 | 22–38 | 18 | 30 |
| 33 | TNAU 101S | 102 | 77 | 90 | 11–24 | 20–38 | 17.5 | 29 |
| 34 | TNAU 102S | 97 | 72 | 85 | 11–24 | 20–38 | 17.5 | 29 |
| 35 | TNAU 103S | 97 | 72 | 85 | 11–24 | 20–38 | 17.5 | 29 |
| 36 | TNAU 106S | 112 | 87 | 100 | 11–24 | 20–38 | 17.5 | 29 |
| 37 | TNAU 107S | 152 | 127 | 140 | 12–21 | 22–35 | 16.5 | 28.5 |
| 38 | TNAU 111S | 152 | 127 | 140 | 12–21 | 22–35 | 16.5 | 28.5 |
| 39 | TNAU 112S | 62 | 37 | 50 | 8–23 | 17–36 | 15.5 | 26.5 |
| 40 | TNAU 113S | 108 | 83 | 96 | 11–23 | 20–38 | 17 | 29 |

| S. NO. | Entry | DFF | DFBD | DPMC | MIN–MAX GDL | MIN–MAX CBE | CFT | CST |
|---|---|---|---|---|---|---|---|---|
| 41 | TNAU 114S | 112 | 87 | 100 | 11–23 | 20–38 | 17 | 29 |
| 42 | TNAU 115S | 102 | 77 | 90 | 11–24 | 20–38 | 17.5 | 29 |
| 43 | TNAU 115S-1 | 119 | 94 | 107 | 12–24 | 22–38 | 18 | 30 |
| 44 | TNAU 116S | 111 | 86 | 99 | 11–24 | 20–38 | 17.5 | 29 |
| 45 | TNAU 120S | 102 | 77 | 90 | 11–23 | 20–38 | 17 | 29 |
| 46 | TNAU 126S-1 | 101 | 76 | 89 | 11–24 | 20–38 | 17.5 | 29 |
| 47 | TNAU 126S-2 | 108 | 83 | 96 | 11–24 | 20–38 | 17.5 | 29 |
| 48 | TNAU 127S | 112 | 87 | 100 | 11–24 | 20–38 | 17.5 | 29 |
| 49 | TNAU 129S | 99 | 74 | 87 | 11–24 | 20–38 | 17.5 | 29 |
| 50 | TNAU 131S | 98 | 73 | 86 | 11–24 | 20–38 | 17.5 | 29 |
| 51 | TNAU 132S | 108 | 83 | 96 | 11–24 | 20–38 | 17.5 | 29 |
| 52 | TNAU 135S | 118 | 93 | 106 | 12–24 | 22–38 | 18 | 30 |
| 53 | TNAU 136S | 94 | 69 | 82 | 11–24 | 20–38 | 17.5 | 29 |
| 54 | TNAU 137S-1 | 110 | 85 | 98 | 11–24 | 20–38 | 17.5 | 29 |
| 55 | TNAU 137S-2 | 108 | 83 | 96 | 11–24 | 20–38 | 17.5 | 29 |
| 56 | TNAU 142S | 102 | 77 | 90 | 11–24 | 20–38 | 17.5 | 29 |
| 57 | TNAU 143S | 102 | 77 | 90 | 11–24 | 20–38 | 17.5 | 29 |

**Notes.**

CST, Critical sterility temperature in °C; CFT, Critical fertility temperature in °C; DFF, Days to fifty percent flowering; DFDB, Days to first bract differentiation; DPMC, Days to pollen mother cell formation; GDL, Gudalur (fertility favoring location); CBE, Coimbatore (Sterility favoring location).

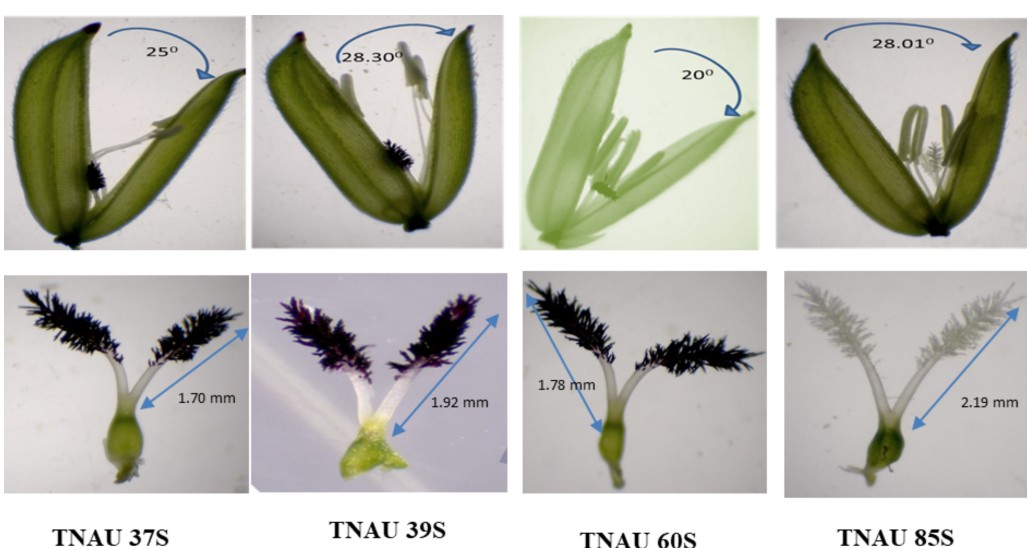

**Figure 4** **Characteristics of stigma and angle of glume opening in the best TGMS lines.**

of *tms8* and *tms10* was the most commonly observed across the lines. Notably, two lines, TNAU 60S and TNAU 38S, contained all four tested genes. Among the lines carrying three TGMS genes, two distinct combinations were identified. TNAU 59S-1, TNAU 82S,
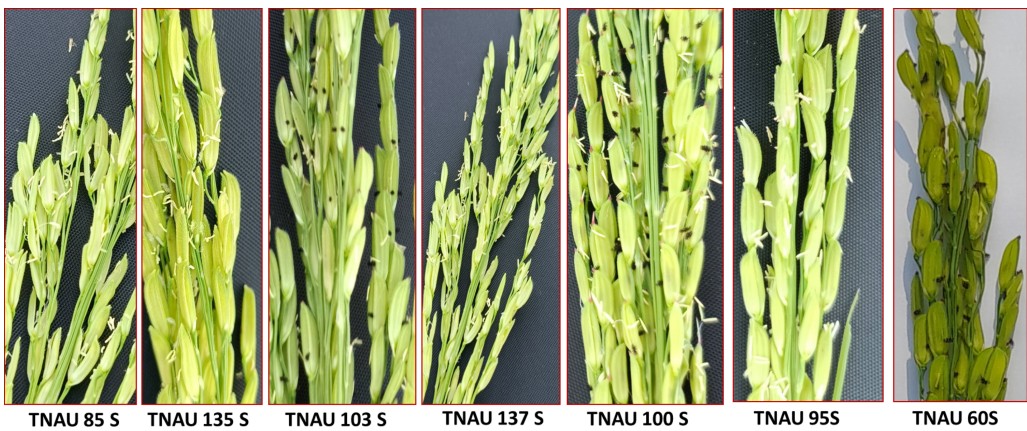

| TNAU 85 S | TNAU 135 S | TNAU 103 S | TNAU 137 S | TNAU 100 S | TNAU 95S | TNAU 60S |

**Figure 5** Stigma exertion comparison of promising TGMS lines.

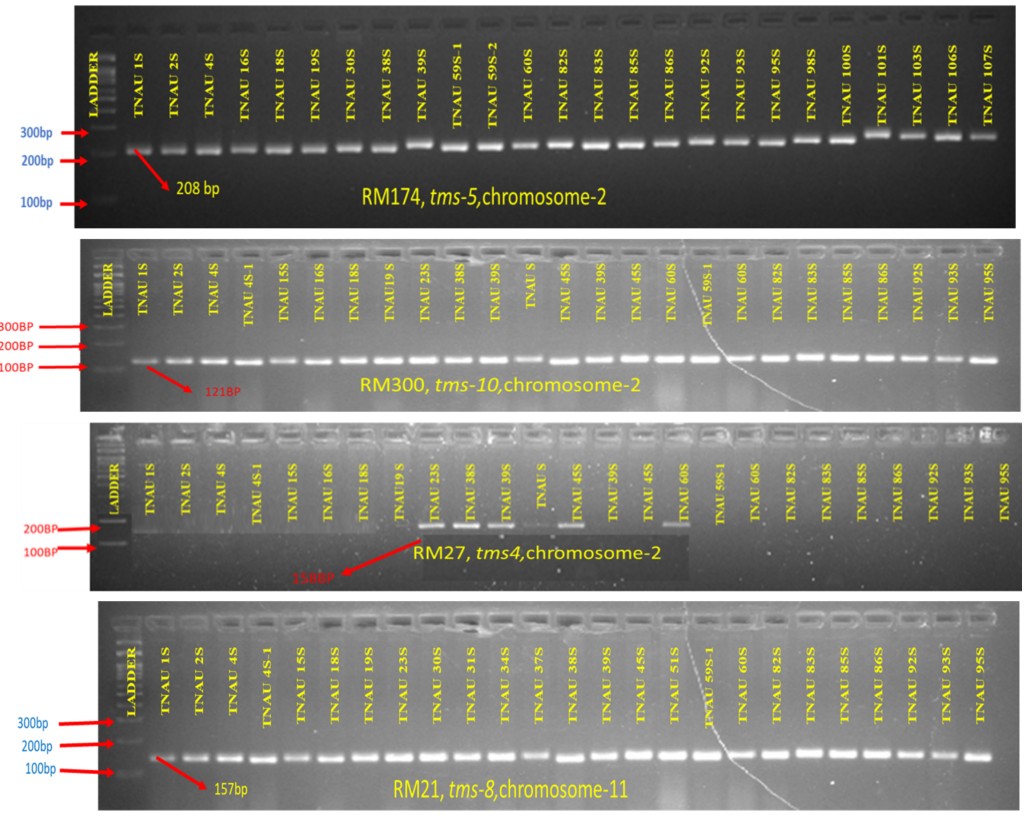

**Figure 6** Molecular profiling of TGMS lines for different *tms* genes (RM174-tms5; RM300-tms10; RM27-*tms4*; RM21-*tms8*).

TNAU 83S and TNAU 85S carried *tms5, tms8,* and *tms10,* while TNAU 23S and TNAU 39S contained *tms4, tms8* and *tms10.* Additionally, *tms8* and *tms10* were found together in 18 lines, reflecting their strong co-occurrence. The *tms10* gene was observed alone in four lines,

**Table 4  Molecular profiling of TGMS lines.**

| S. No. | Gene combination | List of genotypes |
|---|---|---|
| 1 | *tms8* | TNAU 95S, TNAU 100S, TNAU 112S, TNAU 113S, TNAU 114S, TNAU 136S, TNAU 4S, TNAU 4S-1 |
| 2 | *tms10* | TNAU 59S-2, TNAU 71S, TNAU 120S, TNAU 126S-1 |
| 3 | *tms8, tms10* | TNAU 1S, TNAU 2S, TNAU 15S, TNAU 18S, TNAU 19S, TNAU 30S, TNAU 31S, TNAU 34S, TNAU 37S, TNAU 51S, TNAU 98S, TNAU 106S, TNAU 107S, TNAU 111S, TNAU 131S, TNAU 137S-2, TNAU 142S, TNAU 143S |
| 4 | *tms5, tms8* | TNAU 86S, TNAU 92S, TNAU 93S |
| 5 | *tms4, tms8* | TNAU 45S |
| 6 | *tms5, tms8, tms10* | TNAU 59S-1, TNAU 82S, TNAU 83S, TNAU 85S |
| 7 | *tms4, tms8, tms10* | TNAU 23S, TNAU 39S |
| 8 | *tms4, tms5, tms8, tms10* | TNAU38S, TNAU 60S |

while the *tms8* gene was present independently in eight lines. Lines carrying multiple gene combinations from diverse genetic backgrounds exhibited robust and effective sterility, demonstrating the significance of gene interactions in controlling sterility expression. These findings emphasize the genetic diversity and adaptability of TGMS lines, highlighting the potential of specific gene combinations, particularly those involving *tms8* and *tms10,* to enhance sterility stability for hybrid rice breeding programs.

## Correlation of male sterility with floral characters and its implications for enhanced outcrossing in hybrid seed production in Coimbatore

The TGMS lines studied here reveal their deep connection to the natural cycles of fertility and sterility, offering a profound potential to harmonize with breeding programs aimed at male sterility and enhanced cross-pollination traits for hybrid seed production. Among them, TNAU 23S and TNAU 39S, carrying the *tms4, tms8* and *tms10* gene combination, demonstrated complete pollen sterility (100%) and strong panicle exertion, measured at 85.1% and 76.2%, respectively (Table S5). These lines also exhibited favourable floral traits, with stigma lengths of 2.1 mm and 1.92 mm, stigma exertion of 20% and 70.8%, and glume angles of 16.7° and 25° respectively. Their single plant yields, recorded at 18.7 g and 16.5 g, reflect their resilience and adaptability. This combination of sterility and robust panicle exertion helps maintain the natural balance by effectively preventing self-pollination while encouraging outcrossing. TNAU 38S and TNAU 60S, which carry the *tms4, tms5, tms8* and *tms10* gene combination, similarly aligned with nature's design for hybrid seed production. Both lines expressed complete pollen sterility (100%) and exhibited floral traits enhancing outcrossing. TNAU 38S achieved 79.4% panicle exertion, with a stigma length of 1.93 mm, stigma exertion of 43.6%, glume ∠ of 11.7° and single plant yield of 21.2 g. TNAU 60S, with 77.3% panicle exertion, stigma length of 1.78 mm, stigma exertion of 41.1%, glume ∠ of 20°, and single plant yield of 31.3 g, stands as a testament to the harmonious integration of genetic diversity and environmental responsiveness. These lines embody the essence of consistent sterility, ensuring the balance needed for controlled environments.

TNAU 59S-1, TNAU 83S and TNAU 85S, with the *tms5, tms8* and *tms10* gene combination, demonstrated harmony between yield potential and floral characteristics. TNAU 59S-1 recorded a yield of 32 g, glume ∠ of 11.7°, stigma length of 1.49 mm, stigma exertion of 49.7%, complete pollen sterility and 78.6% panicle exertion. TNAU 83S showed a yield of 22.7 g, glume ∠ of 18.3°, stigma length of 2.05 mm, stigma exertion of 51.7% and 81.4% panicle exertion, balanced by 100% pollen sterility. Similarly, TNAU 85S achieved a yield of 19 g, glume ∠ of 28.01°, stigma length of 2.19 mm, stigma exertion of 75%, and 75.1% panicle exertion, maintaining complete pollen sterility. The wider glume angles and higher stigma exertion percentages in these lines echo nature's intent, facilitating cross-pollination and enabling the union of diverse genetic material. By carrying the *tms5, tms8* and *tms10* combination, these lines reflect an evolutionary harmony that enhances hybridization potential, aligning with the cycles of nature and the needs of humanity. These findings underscore the sacred interdependence between genetic diversity and environmental interaction, providing a path forward for sustainable and adaptive hybrid rice breeding. TGMS genotypes ranked by composite scores highlight TNAU 85S as the top performer with balanced traits: 75% panicle emergence, 2.19 mm stigma length, and 28.01° glume angle (score: 0.8). TNAU 39S ranks second (PE: 76.2%, score: 0.67) due to shorter stigma length (1.92 mm) and single plant yield (16.5 g). TNAU 83S achieves the highest PE (81.4%) but lower stigma exertion (51.7%) places it third (score: 0.58). Composite scoring effectively identifies genotypes for enhancing outcrossing efficiency (Fig. 7).

## Correlation analysis of genetic markers and pollen sterility patterns in TGMS rice lines

The analysis of genetic markers in TGMS rice lines reveals a dynamic interplay between specific allele combinations and pollen sterility as shown in the Table S4. Certain lines, such as TNAU 1S, TNAU 2S and TNAU 4S, exhibit 100% sterility associated with uniform allelic patterns across markers RM174, RM5897, RM6247, RM7575, RM21, RM224, and RM300. These patterns suggest that the cumulative contribution of these markers aligns with complete sterility, making them valuable tools for selecting lines with strong sterility potential. This alignment highlights the harmony within the genetic network, reinforcing the connection between marker combinations and sterility outcomes. However, the journey toward sterility in some lines deviates from this uniformity, reflecting the broader complexity of genetic and environmental interactions. Lines such as TNAU 30S (8.2% sterility), TNAU 82S (24.3% sterility), TNAU 86S (23.4% sterility) and TNAU 93S (34.8% sterility) illustrate that additional forces whether unexamined genetic elements, environmental conditions or epistatic interactions contribute to their incomplete sterility. These variations emphasize the interconnectedness of genetic factors and environmental cycles, suggesting that sterility arises not from isolated markers but from a symphony of interacting elements.

Interestingly, near-complete sterility is observed in lines such as TNAU 53S (93.4% sterility), TNAU 114S (92.8% sterility), TNAU 115S (92.8% sterility), TNAU 135S (95.5% sterility) and TNAU 136S (91.4% sterility). Although these lines share allelic patterns with fully sterile lines, they do not achieve complete sterility. This distinction suggests that subtle

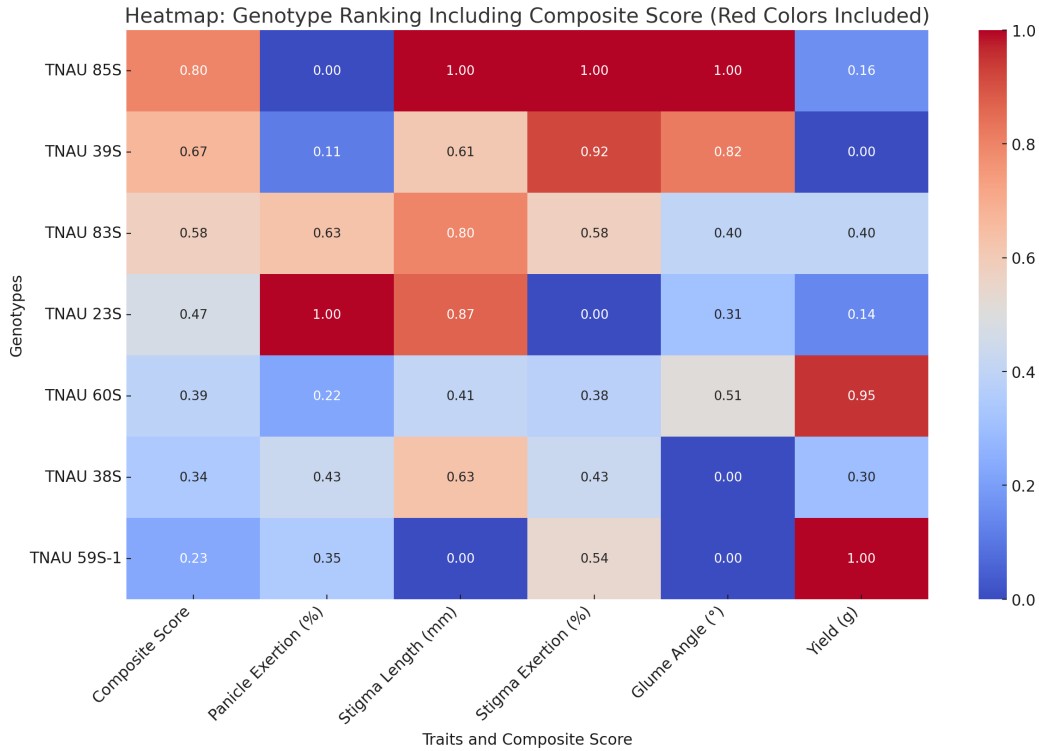

**Figure 7  Ranking of TGMS genotypes based on trait-based composite scores for outcrossing efficiency in rice.**

variations whether minor genetic differences, incomplete gene penetrance or environmental interactions disrupt the balance needed for absolute sterility. Such outcomes reflect the adaptive harmony between genetic potential and environmental forces, underscoring nature's capacity for variation within broader patterns of order. Further adding to this complexity, lines such as TNAU 59S-2, TNAU 98S and TNAU 143S achieve 100% sterility despite having a "0" allele at marker RM21, which is typically associated with sterility. This observation suggests that RM21's variation may play a lesser role in sterility under specific genetic or environmental contexts, highlighting the flexibility and resilience of sterility mechanisms. Conversely, TNAU 113S, with only 11.7% sterility despite a similar allelic pattern to highly sterile lines, underscores the influence of additional genetic interactions or environmental factors that suppress sterility. These exceptions reflect the intricate web of connections that govern sterility, where each thread influences the whole.

## DISCUSSION

The TGMS lines developed at Tamil Nadu Agricultural University reflect a journey of transformation, evolving from lines with poor floral traits, early maturity and limited stability to robust genotypes enriched through the integration of elite varieties. In hybrid rice breeding, quality now walks alongside yield as a primary objective. Meeting diverse consumer preferences remains a challenge, as it depends on the harmony between regional

needs and genetic diversity. This study evaluated 57 TGMS lines with grain types ranging from long-bold to short-slender, illustrating their potential for developing hybrids that cater to varied consumer demands. The pollen sterility of most lines reached 100% under sterility-inducing conditions at Coimbatore, while fertility at Gudalur ranged from 31.9% to 98.6%. The critical fertility temperature (CFT) ranged from 15.5 °C to 19 °C at Gudalur, while the critical sterility temperature (CST) ranged from 26.5 °C to 31 °C at Coimbatore. These thresholds, classified as low CFP (<24 °C) and low CSP (<32 °C), align with the environmental cycles of subtropical regions, making them ideal for hybrid seed production in countries like India (*Kannaiyan et al., 2002*). Lines such as TNAU 60S and TNAU 38S exhibited an extended sterility period of six months (March to August), a trait that mirrors the cycles of the land and provides significant advantages for hybrid seed production, as also reported by *Kanimozhi et al. (2018)*.

Among the 57 lines, TNAU 15S and TNAU 112S were extra-early flowering types, making them valuable for hybrids adapted to drought-prone areas. TNAU 93S and TNAU 16S, with early flowering, dwarf stature and medium bold grains, are promising donors for short-duration, semi-dwarf hybrids that withstand lodging. The high productivity of TNAU 16S is evident from its maximum number of productive tillers, highlighting its strong yield potential. Lines such as TNAU 126S-1 and TNAU 126S-2 exhibited greater panicle exertion and wider glume-opening angles, traits essential for improved seed yield in hybrid production. These findings align with *Srimathi et al. (2019)*, who documented the positive correlation of panicle exertion and glume-opening angle with seed yield. TNAU 95S, TNAU 16S, TNAU 19S, TNAU 135S, TNAU 126S-1 and TNAU 126S-2 demonstrated consistent sterility under sterility-inducing conditions and high pollen and spikelet fertility at fertility-inducing conditions. TNAU 19S stood out with more than 70% stigma exertion, highlighting its suitability for hybrids that require enhanced outcrossing. Studies by *Roy & Kumaresan (2019)* and *Kanimozhi et al. (2018)* emphasized the importance of high stigma exertion and wide glume angles for improving outcrossing efficiency. Similarly, *Manonmani, Pushpam & Robin (2016)* demonstrated the stable sterility of lines such as TNAU 45S, TNAU 60S, TNAU 95S, TNAU 19S and TNAU 39S under two sterility-inducing environments.

TNAU 114S, TNAU 131S, TNAU 34S, TNAU 37S and TNAU 113S showed high single-plant yields, with TNAU 114S achieving the highest grain count per panicle. While TNAU 34S featured long slender grains, TNAU 114S and TNAU 113S, derived from the cross TNAU 4S/BPT5204, lacked complete sterility. The partial restoring ability of BPT5204, noted by *Ponnuswamy et al. (2020)*, suggests its potential as a male parent for two-line hybrid production rather than TGMS line development. TGMS is governed by recessive nuclear genes. Over time, genes such as *tms1, tms2, tms4, tms5, tms8* and *tms10* have been identified (*Ali & Wani, 2021*). Among these, *tms8* was the most extensively dispersed, present in over 50% of genotypes. TNAU 38S and TNAU 60S carried all four genes (*tms4, tms5, tms8* and *tms10*), highlighting their adaptability. Modern tools like CRISPR/Cas9 technology have further enhanced TGMS breeding, as seen in *tms5* mutants (*Barman et al., 2019*). The TGMS line TNAU 60S, with its extended sterility period and 100% sterility, was instrumental in developing South India's first medium slender two-line

hybrid, TNTRH 55 (*Manonmani et al., 2012*). Lines like TNAU 59S-1 and TNAU 85S, carrying the *tms5, tms8* and *tms10* gene combination, achieved a balance between yield, panicle exertion, stigma exertion and sterility, making them ideal candidates for breeding programs.

In summary, TGMS lines with *tms4, tms8, tms10,* or *tms4, tms5, tms8, tms10* combinations offer ideal solutions for complete sterility, while lines with *tms5, tms8,* and *tms10* balance sterility with agronomic performance. These lines, guided by nature's cycles and human innovation, represent a harmonious convergence of genetics, environment, and agricultural potential.

## CONCLUSIONS

In conclusion, the findings reveal that while specific allele combinations across markers RM27, RM174, RM5897, RM6247, RM7575, RM21, RM224 and RM300 strongly align with pollen sterility, variations in sterility percentages among certain lines (*e.g.*, TNAU 113S, TNAU 82S, TNAU 86S) highlight the presence of deeper genetic and environmental interactions. This study emphasizes the balance between sterility and fertility, showcasing the interdependence of genetic diversity, environmental responsiveness and epigenetic factors.

### Funding
This work was funded by the ICAR-"Consortium Research Platform (CRP) on Hybrid Technology: RICE" and Department of Rice, Tamil Nadu Agricultural University". The funders had no role in study design, data collection and analysis, decision to publish, or preparation of the manuscript.

### Grant Disclosures
The following grant information was disclosed by the authors:
ICAR-"Consortium Research Platform (CRP) on Hybrid Technology: RICE".
Department of Rice, Tamil Nadu Agricultural University.

### Competing Interests
The authors declare there are no competing interests.

### Author Contributions
- Nagendra Naidu B conceived and designed the experiments, performed the experiments, analyzed the data, prepared figures and/or tables, authored or reviewed drafts of the article, and approved the final draft.
- Manonmani Swaminathan conceived and designed the experiments, performed the experiments, analyzed the data, prepared figures and/or tables, authored or reviewed drafts of the article, and approved the final draft.

- Nivedha Rakkimuthu conceived and designed the experiments, performed the experiments, analyzed the data, prepared figures and/or tables, authored or reviewed drafts of the article, and approved the final draft.
- Pushpam Ramamoorthy conceived and designed the experiments, performed the experiments, analyzed the data, prepared figures and/or tables, authored or reviewed drafts of the article, and approved the final draft.
- Kumaresan Dharmalingam conceived and designed the experiments, performed the experiments, analyzed the data, prepared figures and/or tables, and approved the final draft.
- Raveendran Muthurajan conceived and designed the experiments, performed the experiments, analyzed the data, prepared figures and/or tables, authored or reviewed drafts of the article, and approved the final draft.
- Selvi Duraisamy conceived and designed the experiments, performed the experiments, analyzed the data, prepared figures and/or tables, authored or reviewed drafts of the article, and approved the final draft.
- Tushar Arun Mohanty conceived and designed the experiments, performed the experiments, analyzed the data, authored or reviewed drafts of the article, and approved the final draft.
- Bonipas Antony John conceived and designed the experiments, performed the experiments, analyzed the data, prepared figures and/or tables, and approved the final draft.

## Data Availability

The raw data is available in the Supplemental Files.

## Supplemental Information

Supplemental information for this article can be found online at http://dx.doi.org/10.7717/peerj.18803#supplemental-information.

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
