# Peer review of "Molecular profiling and phenotypic evaluation of thermo-sensitive genic male sterility genes for high-yielding rice hybrids (Oryza sativa L.)"

_PeerJ, doi:10.7717/peerj.18803_

## Round 0.1 · original submission · Major Revisions

Your manuscript requires major revisions that should address the comments from the reviewers.

Reviewer 1 ·

Basic reporting

Basic Comments:
The manuscript presents a comprehensive analysis of TGMS lines, combining phenotypic data with molecular profiling. However, the language needs huge improvement in terms of clarity, grammar and coherence. The manuscript occasionally shifts between past and present tense, which can be confusing. For example, "rice is an essential staple crop" (present tense) followed by "investigation of the stability" (implied past tense) creates inconsistency. Maintaining a consistent tense throughout the manuscript would improve readability. There are few spelling and punctuation errors that need correction. For example, "enhacing" should be "enhancing," and "It is" in the first sentence should be lowercased or rephrased. Please carefully proofread the manuscript to correct grammatical errors, tense inconsistencies, and spelling mistakes.
Sentences are often lengthy and complex, which may obscure the intended meaning. Breaking down complex sentences into shorter, more direct ones would improve readability. The manuscript lacks coherence, with some ideas presented abruptly without sufficient details. For example, in introduction section, the transition between discussing hybrid rice technology and male sterility systems could be smoother, providing context and connection between the topics. Please Improve Clarity and Readability of the manuscript. There are several grammatical issues that need correction. For example, in the first sentence, "which It is crucial" should be corrected to "which is crucial." Additionally, "investigation of the stability and variance in quality" is an incomplete sentence and should be revised for clarity and completeness. The manuscript sometimes uses overly complex or technical terms without explanation, which could alienate readers who are not specialists in the field. Consider explaining technical terms or using simpler language where appropriate. The technical content appears to be accurate, but it could benefit from more precise definitions and explanations of key concepts.
Abstract: Lacking key values. Please add key values in this section. Some sentences are not clear or misleading.
Keywords: Seems ok.
Introduction: The introduction should clearly define the scope of the manuscript and outline the main objectives. While the current introduction mentions the importance of hybrid rice, it could be more focused on the main theme of the manuscript. The manuscript would benefit from a clearer structure, with each section logically leading to the next. Please clearly delineate different topics, such as "Hybrid Rice Technology," "Male Sterility Systems," “Cytoplasmic Male Sterility (CMS), “Environment-sensitive Genic Male Sterility (EGMS) systems”, "Thermo-sensitive Genic Male Sterility," and "Marker-Assisted Selection." For example, the description of the. Please Provide more detailed explanations of the key concepts, and how these systems work at the molecular level? The section on the EGMS system could include more information on the molecular basis of temperature sensitivity and how this affects gene expression and mechanisms discussed in the manuscript.
Please provide the novelty of the study and clear rational supported by latest literature.
Materials and Methods: The section must provide an adequate amount of detail for replicability. When referencing protocols (e.g., "according to the protocol given by Vinodhini et al., 2019"), it might be helpful to briefly describe the protocol or its critical steps, especially if it is not commonly known. The authors must add certain technical details, such as the exact amount, timing and types of fertilizers, exact protocol including the environmental conditions and the concentrations of solutions used in DNA extraction. Given the study's scope, involving 57 rice TGMS lines, the application of rigorous statistical analysis is essential. Without the use of appropriate statistical methods, the predictive power and interpretability of the experimental results remain uncertain, potentially compromising the validity of the conclusions drawn.

In Results section to enhance clarity, the authors should provide clearer context for the significance of the findings. Given that temperature is a critical factor in the study of TGMS lines, it is imperative that the authors provide a comprehensive analysis and description of the environmental data depicted in Fig. 1 and temperature induced critical sterility and fertility in Fig. 2 must be explained. This data should be thoroughly discussed to accurately contextualize the results and highlight the influence of temperature on the observed outcomes.
In Fig. 3a: 1st band of TNAU185 is curved? Why? Please provide the clear and correct image. If you have good bands with prominent results then use those images.
Fig. 3 Some of the bands are missing. Please provide their original source file.
In Table 3, the authors have reported errors and have added asterisks to certain values. However, the materials and methods section lack any mention of the statistical methods or software used in the analysis. It is essential to explicitly clarify whether the term "Errors" refers to the standard error of the mean or another statistical metric. Additionally, the station where the parameters in Table 3 were measured has not been specified. This information is crucial for understanding the environmental context of the data.
Furthermore, Table 4 presents data from the Gudalur station. It would be informative to include a comparison or at least mention the corresponding parameters for the 57 lines evaluated at the Coimbatore station, as this would provide a more comprehensive understanding of the results across different environmental conditions.
Please clearly specify in Table 5 whether the Critical Sterility and Fertility Temperatures of TGMS lines are consistent across both stations or specific to one.
In Table 6, clearly indicate which station's environment is inducing fertility and which is inducing sterility for the TGMS lines.
Captions of all Figures and Tables are not correct. Also mention in the Captions that what does Fig. 1a, 1b, are indicating? Also include complete forms of all abbreviations in the captions.
In Discussion: There is some inconsistency in the use of abbreviations and gene nomenclature (e.g., "TGMS" is sometimes written as "tgms"). Ensuring uniformity in these terms throughout the manuscript will enhance professionalism and reduce confusion. The discussion could benefit from more critical analysis of the results, especially in terms of the implications for breeding programs. Currently, the manuscript reports findings but does not contextualize them from the previous literature. The manuscript cites relevant literature, but some statements would be strengthened by additional latest citations, particularly when discussing well-established concepts like the role of specific TGMS genes.
The conclusion section could be enhanced by refining the language and providing additional context for some of the findings.
The references used in the manuscript are appropriate, but they need to be formatted consistently. Ensure that all references are presented in the correct format and that in-text citations match the reference list. Additionally, some claims in the manuscript could be strengthened by citing more recent research to support the statements. Please Ensure that all references are correctly formatted and consider adding more recent citations to support the claims made in the manuscript. References should follow the Peer J reference style including DOI.

Specific Comments
Line No. 51-52: which It is crucial for ensuring global food security (1). Investigation of the stability and variance in quality of hybrid rice in various conditions. What does authors intendsto convey is not clear from these sentences.

Line No. 53 Under constant irrigation circumstances, yields of rice have grown by 20%.30% thanks to hybrid technology? Enhance the clarity here what is the link of irrigation and hybrid seed technology?

Line No. 58-59: First noted in pepper by Martin and Crawford in 1951 (3), this male sterility mechanism was later found in other crops. Please provide suitable reference here?

Line No. 99: The experimental materials comprised of 57 rice TGMS lines..." should be "The experimental materials comprised 57 rice TGMS lines..." (remove "of").
Line No. 115: There is occasional use of overly complex phrases that could be simplified without losing meaning, such as "The observations were documented from randomly selected five individual plants for each genotype and averaged to obtain mean values." This could be simplified to, "Observations were recorded from five randomly selected plants per genotype and averaged."
Line No. 119: "Three millimetre tungsten beads were inserted to each of the 1.5 ml microtubes..." should be "Three-millimetre tungsten beads were inserted into each of the 1.5 ml microtubes..." (correct preposition and hyphenation).
Line No. 122: tissues were broken up and homogenised into a fine powder." could be more precisely written as "the tissues were homogenized into a fine powder."
Line No. 139: Higher yields in hybrid seed production depends on out crossing rate..." should be "Higher yields in hybrid seed production depend on the outcrossing rate..."
Line No. 160: The critical fertility temperature ranged from 15 - 170C..." should be "The critical fertility temperature ranged from 15 to 17°C..."
Line No. 190: Two accessions viz., TNAU 38S and TNAU 60S shown positive amplification..." should be "Two accessions, viz., TNAU 38S and TNAU 60S, showed positive amplification..."
Some sentences are overly complex and could be simplified for better readability. For instance, In Line 202: "Effective complete sterility was shown by the lines with the multiple gene in different combinations in different genetic backgrounds." could be rephrased as "Lines with multiple gene combinations in different genetic backgrounds exhibited effective complete sterility."
Line 278-279: The presence of more genes ensures stability of these lines at sterile phase," it might be clearer to write "The presence of multiple genes ensures the stability of these lines during the sterile phase."
The use of technical terminology is appropriate for the target audience, but some terms could be better defined for clarity. For instance, terms like "panicle exertion" and "glume angle" are used without explanation, which could be problematic for readers unfamiliar with these specific traits.
In Line 286-287: the phrase "s considered ideal" likely contains a typographical error and should be "is considered ideal."
The sentence in Line No. 292-293 "Promising lines which exhibiting all the major genes" is awkwardly phrased and should be revised to "Promising lines exhibiting all the major genes."

Experimental design

Experimental design seems okay. But needs more clarity regarding the number of replications used as it is essential to judge the validity of the findings. The manuscript in its current form lacking the mentioning of statistical tools used to interpret the results. Moreover, inter-treatment variation can be traced by suitable post-hoc test as HSD, LSD etc. that are used for applying appropriate annotations or letters to indicated statistically significant differences between treatments.

Validity of the findings

The findings can be promising provided the authors provide complete details, and make corrections as suggested in the basic reporting for improving the validity of the findings, enhancing clarity and reproducibility.

Additional comments

By addressing mentioned gaps, language and grammar issues the article could be technically sound, and appealing to the scientific community, potentially increasing its impact and citation rates.

Reviewer 2 ·

Basic reporting

The authors provided insights into the hybrid rice breeding system by conducting studies on Molecular Profiling and Phenotypic Evaluation of Thermo-genetic Male Sterility Genes for High-Yielding Rice Hybrids (Oryza sativa L.) at two different locations in India, with 462 and 133 meters above sea level. The study is a nice attempt and utilizes two natural conditions with varying temperatures for thermal sensitivity in inducing male sterility/fertility in rice.

However, there are a few concerns, which should be addressed before the article can meet the publishing standards of the journal. For Example, the article is missing logical coherency in its write-up. For instance, the Introduction primarily focuses on aspects such as plant height and the role of Thermo-genetic Male Sterility (TGMS) lines. In contrast, the Results section places significant emphasis on traits like flowering and the number of tillers. This disconnection creates a fragmented narrative and diminishes the article’s overall appeal. It appears as though different sections may have been written by different contributors, resulting in a disjointed and unfocused presentation.
Secondly, run-on sentences are noted, where one sentence comprises 3-4 lines, with more than two ideas embedded. This requires reading back and forth a couple of times before one can understand it. Simply the authors are suggested to split such sentences into small understandable statements/sentences.
Following problems were noted in th ms and should be corrected as suggested:
Line 29: "found to have very productive tillers" remove 'very'
Line 51: increased rice output, which It is crucial for ensuring remove 'It'
Line 51: investigation of the stability and variance in quality of hybrid rice in various conditions. Incomplete sentence or does not makes any sense where it is inserted.
Line 157: 13 0C - 380C It looks like authos are using 0 and superscripting it, however, a symbol should be inserted, Secondly space should be there between number and degree C
Line 267: they are non-allelic to any known tgms genes replacle tgms with TGMS
Line 299: Tamil nadu agricultural university, first letter Capital for the words
etc......

Experimental design

It is unclear how 57 male sterile lines were selected for the Experiment.

The current methodology does not clearly explain if similar traits were noted and compared at both locations i.e. Gudalur and Coimbatore. Needs a better and clearer stance as both locations have significant temperature differences which on the one hand induces either male sterility or fertility, on the other hand, the temperature has a dictating role in almost all the growth factors.

The article lacks a detailed description of key morphological parameters essential for evaluating male sterility. Important aspects such as pollen fertility, sterility, stigma length, and glume opening angle are not addressed, which makes this section incomplete and less informative.

The Abstract fails to adequately convey the significance of male sterility in hybrid rice production. It should be revised to highlight the importance of male sterility in the context of hybrid rice development and its implications for breeding practices.

Validity of the findings

The manuscript contains poorly used abbreviations, some of which cannot be traced throughout the text. For example, the abbreviation CSP and its relation to CFP, RM, and other terms need clarification and consistent usage throughout the manuscript. The authors are suggested to please write the complete term before writing its abbreviation. Afterward, don’t write the whole term in a section. Maintain uniformity.
In summary, to enhance the article’s quality and coherence, the authors should focus on integrating the various sections into a unified narrative that consistently addresses the central theme of male sterility in hybrid rice production. Emphasizing the relevance of critical traits and providing a thorough description of key morphological parameters will significantly improve the manuscript's clarity and impact.
The references cited in the manuscript are generally appropriate; however, they need to be consistently formatted. Please ensure that all references adhere to the correct format and that in-text citations are accurately reflected in the reference list. Additionally, some assertions in the manuscript could be bolstered by incorporating more recent research to support these statements. Make sure all references are correctly formatted according to the PeerJ reference style, including the inclusion of DOIs, and consider adding more up-to-date citations to reinforce the claims made.

Reviewer 3 ·

Basic reporting

The abstract should be a brief summary of the study conducted. It should include a concise overview of the introduction, materials and methods, findings, and conclusions. We do not see such an abstract

The citation format of the PEERJ journal is not by number but in the form of author et al. (year ). All instances of this need to be changed throughout the entire text.
Investigation 52 of the stability and variance in quality of hybrid rice in various conditions. This sentence is irrelevant.
In the introduction paragraph, the text was not written with a logical connection between sentences. Irrelevant and disconnected sentences follow each other. This needs to be revised. Additionally, there should not be sharp transitions between paragraphs in terms of meaning.
Line80-87. There is no reference.
Under the 'Plant Material' heading, the method was provided. Either provide information only about the plants or use a heading like 'Plant Material and Method.
There is no space between passages.
Line 126-137. Please give a reference about PCR protocol.
The discussion section should be enriched. İt is too weak..

Experimental design

-

Validity of the findings

-

Additional comments

Aside from minor deficiencies and a weak discussion section, an important topic was addressed and well-structured.

---

## Round 0.2 · Major Revisions

It seems that you have not sufficiently addressed the concerns of the reviewers. Your manuscript still requires major revisions. However, I must state once again that failure to make revisions and the failure to improve your manuscript will result in its rejection. In addition to the reviewers' recommendations, please consider the following points.

-Write percentage instead of percentage throughout the text.
-It would be appropriate to include an assessment of the polymorphism and efficiency of the primers. In addition, it is necessary to show the genetic diversity of the genotypes with a dendogram. If your aim is only to detect the presence of that allele, then you should comment by correlating them with your morphological data. The molecular part of the study is written in a inexperienced manner. Please make it more explanatory.

Reviewer 1 ·

Basic reporting

I am pleased to see that authors answered most of the queries but they must add the information or make desired corrections regarding the following previous queries
“The minimum and maximum temperatures at the critical stage for sterility and fertility expression were recorded according to the protocol given by Vinodhini et al., (2019) by exposing the lines with the varies environmental changes from the 130C to 380C. The mean minimum and mean maximum temperatures where transition from the fertility to sterility is considered as critical fertility point and critical sterility point respectively. All the observations were recorded from the randomly selected plants per genotype and averaged.” The details provided in this section (Lines 113-119) are a repetition of previously stated information (Lines 104-109)."
“The aqueous phase was then transferred to a fresh tub” Clarify tub or tube? (Lines 126-127)
Finally, gel electrophoresis or a spectrophotometer was used to evaluate the quantity and quality of the DNA. Which technique was used exactly in your work?
Manuscript still contains lot of typographical and grammatical errors. There is inconsistent use of abbreviations.
The fertilizer application and intercultural operations were conducted as per the recommended standard. (93-94). Clearly mention the timing and doses of fertilizers.
Authors did not provide the satisfactory correction to this query. Given the study's scope, involving 57 rice TGMS lines, the application of rigorous statistical analysis is essential. Without the use of appropriate statistical methods, the predictive power and interpretability of the experimental results remain uncertain, potentially compromising the validity of the conclusions drawn.
“In Table 3, the authors have reported errors and have added asterisks to certain values. However, the materials and methods section lack any mention of the statistical methods or software used in the analysis. It is essential to explicitly clarify whether the term "Errors" refers to the standard error of the mean or another statistical metric. Additionally, the station where the parameters in Table 3 were measured has not been specified. This information is crucial for understanding the environmental context of the data.” Regarding this previous query I found nothing related to statistical analysis in the material and method section. Could you mention where did you added this information?
“In Discussion: There is some inconsistency in the use of abbreviations and gene nomenclature (e.g., "TGMS" is sometimes written as "tgms"). Ensuring uniformity in these terms throughout the manuscript will enhance professionalism and reduce confusion.” This problem is still there in Line No. 270.

Experimental design

Experimental design is ok. However, authors should add information regarding the statistical tools used to analyse the results.

Validity of the findings

Findings seems promising.

Reviewer 2 ·

Basic reporting

The article's language is much better and acceptable now.
However, some of the concerns were not clearly understood and hence need clarity.
For example my comment "It is unclear how 57 male sterile lines were selected for the Experiment" meant a description on how the lines were selected:
whenther based on Literature review (Researchers reviewed existing literature on rice male sterility genetics, breeding, and heterosis i.e hybrid efficiency.
OR
Germplasm evaluation: A large germplasm collection was evaluated for desirable traits.
OR
Preliminary screening: Lines were screened for male sterility and grain quality?
OR
Pedigree analysis: Lines' pedigrees were analyzed to ensure genetic diversity.
OR
Molecular marker analysis: Molecular markers were used to confirm genetic relationships.

Experimental design

Following comment was partially discussed. The male sterility is explained but other characters still missing in the materials and methods section. The original comment was:
The article lacks a detailed description of key morphological parameters essential for evaluating male sterility. Important aspects such as pollen fertility, sterility, stigma length, and glume opening angle are not addressed, which makes this section incomplete and less informative.

Validity of the findings

The authors need to work more about the following comment:
In summary, to enhance the article’s quality and coherence, the authors should focus on integrating the various sections into a unified narrative that consistently addresses the central theme of male sterility in hybrid rice production. Emphasizing the relevance of critical traits and providing a thorough description of key morphological parameters will significantly improve the manuscript's clarity and impact.

Reviewer 3 ·

Basic reporting

From my side is ok.

Experimental design

-

Validity of the findings

-

Additional comments

-

---

## Round 0.3 · Minor Revisions

Your manuscript needs further minor revisions

Reviewer 1 ·

Basic reporting

The manuscript still contains typographical and grammatical errors, and at certain points, the writing lacks maturity and clarity. It is strongly recommended that the authors seek assistance from a professional English editing service or a native English speaker.
Line No. 84-89 “Even though TNAU developed TGMS lines, no molecular ……... sterility-inducing and fertility-inducing conditions were used to assess TGMS lines.” These lines are not suitable for materials and methods section. These lines must be added in introduction section if it is lacking this information.
The information which the authors added in rebuttal letter, must be added appropriately in the materials and methods section. “We assessed the DNA quality and quantity using a spectrophotometer by measuring absorbance at 260/280 nm. This ratio provided an estimate of DNA purity, with values close to 1.8 indicating high-quality DNA.”
“We have revised the manuscript to specify the timing and doses of fertilizers, as per the recommended standards provided by the TNAU Agritech Portal (http://www.agritech.tnau.ac.in/), to improve clarity for readers.” Please provide exact dosage of fertilizers, rather than just giving the website link.

Authors have provided this information in their rebuttal letter, but unfortunately it is missing in the manuscript. “Statistical analyses were performed using R software to ensure rigorous evaluation and enhance the accuracy of the results. The "aov" function was utilized for analysis of variance (ANOVA), and the "cor" function was employed for correlation analysis, with a specific focus on the relationships between genetic markers and phenotypic traits. To calculate descriptive statistics, including the mean and standard error of the mean (SEM), the psych package was used, which provides functions for easily computing these metrics.”

The results which are presented under headings “Correlation of Male Sterility with Floral Characters and Its Implications for Enhanced Outcrossing in Hybrid Seed Production in Coimbatore” and “Correlation Analysis of Genetic Markers and Pollen Sterility Patterns in TGMS Rice Lines” are lacking presentation in tables or Figures?
If necessary, RNA was removed by treating the solution with RNase A. If The DNA's quantity and quality were evaluated using a spectrophotometer, including assessment of the optimal 260/280 absorbance ratio of ~1.8. Please correct these sentences (Line No. 133-135).
Line No. 116 “the lines with the varies environmental changes from the 130C to 380C”
correct the degree symbol. Moreover, this symbol is wrongly inserted at multiple places including captions of Figures.
It is recommended to include standard errors alongside the values in Table 6 to provide a clearer representation of variability and enhance the statistical robustness of the data.

Experimental design

Please address the problems identified in basic reporting.

Validity of the findings

Please add standard errors in Table 6, it will help to assess the validity of findings.

Reviewer 2 ·

Basic reporting

The authors have incorporated the suggested changes.

Experimental design

Looks better now.

Validity of the findings

The suggestions included and looks good now.

---

## Round 0.4 · Minor Revisions

Your manuscript has been improved through many revisions. I also found the following deficiencies in my final review. Once you have completed these, your manuscript will be accepted. Please show the file you have made corrections to on the clear file after you have accepted the changes you made in the previous version. This will make my decision process shorter. I won't have to read everything from the beginning, unless you present me with a complicated file.

Line 86: Rabi and Kharif are terms specific to your region. Instead use the globally used terms winter and spring.
Lines 90-98 and 99-104 contain almost the same information. Delete lines 99-104. Write lines 90-98 after the heading "Observations recorded."
Line 118: Write clearly what the extraction buffer is. Also, "or an SDS" is a misspelling. What does this mean? Did you mean to write "CTAB and/or SDS"? Or "CTAB or a SDS"? The prefix "an" is used before words that start with a vowel.
Line 299-306: Why write these here?
The conclusion section should not be a summary of what you have written. Briefly emphasize the clear conclusions you have reached. It is pointless to repeat the same things over and over. Lines 299-306: "In conclusion, the findings reveal that while specific allele combinations across markers RM27, RM174, RM5897, RM6247, RM7575, RM21, RM224, and RM300 strongly align with pollen sterility, variations in sterility percentages among certain lines (e.g., TNAU 113S, TNAU 82S, TNAU 86S) highlight the presence of deeper genetic and environmental interactions. This study emphasizes the balance between sterility and fertility, showing the interdependence of genetic diversity, environmental responsiveness, and epigenetic factors. To fully understand these mechanisms, further exploration of environmental and genetic relationships is needed, guided by the principles of harmony and respect for natureís intricate design." Write these in your conclusion. Delete the existing conclusion section completely.

---

## Round 0.5 · accepted · Accept

Your manuscript includes all changes required for acceptance. Congratulations